# Tumor Necrosis Factor’s Pathway in Crohn’s Disease: Potential for Intervention

**DOI:** 10.3390/ijms221910273

**Published:** 2021-09-24

**Authors:** Cristiano Pagnini, Fabio Cominelli

**Affiliations:** 1Department of Gastroenterology and Digestive Endoscopy, S. Giovanni Addolorata Hospital, 00184 Rome, Italy; 2Digestive Health Research Institute, Case Western Reserve University School of Medicine, Cleveland, OH 10900, USA; fabio.cominelli@uhhospitals.org

**Keywords:** Crohn’s disease, tumor necrosis factor, anti-TNF, infliximab, adalimumab, certolizumab, innate immunity

## Abstract

Crohn’s disease (CD) is a chronic disorder characterized by full thickness patchy inflammation of the gastrointestinal tract. The pathogenesis is multifactorial and involves defective innate immune responses, microbiome alterations, and dysregulated activation of the acquired component of mucosal immunity. One of the molecular mediators that is involved at different levels in the initiation and progression of intestinal inflammation characteristic of CD is tumor necrosis factor (TNF). The present manuscript provides a comprehensive review focused on the potential role of TNF in the different phases of CD pathogenesis, particularly in light of its potential clinical implications. Currently available drugs blocking TNF are evaluated and discussed, specifically for open issues that still remain utilizing such therapy. TNF exerts a paramount role in the established phase of intestinal inflammation that characterizes CD patients, and anti-TNF biologics have definitely changed patient management, offering effective and safe options of treatment. Nonetheless, many patients still do not respond to anti-TNF therapy or experience unwanted side-effects. This could partially be due to the role that TNF plays in intestinal homeostasis that is particularly important during the early phase of the inflammatory process. In fact, emerging evidence supporting the dichotomous role of TNF and the identification of molecular markers will guide a more tailored and refined therapy for CD patients in the near future.

## 1. Introduction

Crohn’s disease (CD) is a complex, chronic disorder of the intestine with still unknown etiology. It is one of the etiopathogenic forms of inflammatory bowel disease (IBD), a group of disorders wherein dysregulated mucosal immune responses causes a persistent inflammation of the gut with a clinical relapsing-recurrent pattern of disease [1]. Unlike ulcerative colitis (UC), the other major form of IBD, in which inflammation is limited histologically to the mucosal surface and localized to the colon, inflammation in CD affects all the intestinal wall layers of the GI tract. CD may involve virtually all areas of the digestive tract with the terminal ileum representing the most frequent location. The natural history of the disease is typically characterized by an initial phase, ‘inflammatory phenotype,’ where the clinical manifestations are mainly related to the mere inflammation of the intestine. This is usually followed, within a variable number of years, with a more complicated disease phase, wherein strictures of the bowel (‘stenosing phenotype’) and fistulas that form between affected intestines and other organs or areas of intestine (‘fistulizing phenotype’) may occur. The symptoms vary according to disease phenotype and status (remission vs. flare), but mainly manifest as abdominal pain, diarrhea, fever, weight loss, and possible occurrence of sub-occlusive symptoms (cramping pain, vomiting, difficult feces/gas evacuation) or other complications (i.e., abscesses, fistulas). Extraintestinal manifestation is not infrequent, and rheumatic and dermatological involvement occurs in up to 40% of patients [2]. CD is a rare disease, but one with increasing frequency worldwide: incidence rates are up to 20.2 per 100,000 persons/year in North America and 12.7 per 100,000 persons/year in Europe. Prevalence rates are also up to 319 per 100,000 persons in North America and 322 per 100,000 persons in Europe [3].

Among the so-called conventional therapies, corticosteroids are still the mainstay of treatment for acute inflammatory flares, and budesonide [4], characterized by a more favorable safety profile, can be used for mild-to-moderate inflammatory flares [5,6]. Immunosuppressive treatments (thiopurines and methotrexate) have been partially scaled down with the advent of biologics but are still indicated for maintenance of remission as steroid sparing agents and in combination with infliximab for immunogenicity reduction [7]. Salicylates, antibiotics, and probiotics may be used in specific situations, but they lack solid evidence for efficacy in CD patients [8,9]. Surgery represents a still frequent therapeutic option for CD patients, and it is generally indicated for medically refractory and complicated (stenosis, fistulas) conditions [10].

In general, the goal of a successful therapeutic approach is induction and maintenance of disease remission; however, the specific definition of “disease remission” consistently has changed throughout the years. The target for successful therapy has changed from simple ‘symptom control’ to the current identification of solid therapeutic endpoints such as clinical and biochemical normalization, mucosal and transmural healing patient reported outcomes (PROs), and quality of life. These are better indicators of complete short- and long-term disease control and are also related to a decreased rate of complications (steroid use, hospitalization, surgery) [11,12,13]. During the last few decades, this specific transition regarding successful outcomes of disease treatment has been accompanied by the development of biologic drugs.

Biologic (or biotechnologic) drugs are large molecules (often monoclonal antibodies) produced from biologic sources (i.e., cell lines). They are characterized by a complex and specific production process, resulting in the generation of compounds with a certain range of structural variability and a defined biological effect [14]. For increasing short- and long-term anti-inflammatory effects, and achieving acceptable safety profiles, these drugs have consistently expanded the limited efficacy of conventional therapies and dramatically changed the approach and management of IBD patients.

In regard to targeted therapy, the knowledge of disease pathogenesis has expanded throughout the years thanks to intense research and progress in laboratory technology, even though substantial pitfalls persist [15]. Genetic predisposition has long been indicated as a potential factor for disease occurrence. Genome-wide association studies (GWAS) have identified nearly 240 genetic risk loci—with almost 30 shared between CD and UC—prevalently associated with genes involved in epithelial barrier function, innate mucosal defense, cell migration, autophagy, and adaptive immunity. The identification and characterization of these loci has contributed to a better understanding of IBD [16,17,18].

Disease pathogenesis of CD has been associated early on with prevalent Th1 immune responses, so that dysregulation of acquired immunity has been indicated as a prevalent mechanism for disease occurrence. Indeed, several experimental observations support a role for Th1 cells in disease pathogenesis, and CD patients show a high prevalence of Th1 cells in the gut lamina propria that produce large amounts of IFNγ [19,20,21,22,23]. Besides increased activity of Th1 cells with a pro-inflammatory phenotype, a reduction of immune cells with regulatory activity has also been associated with CD pathogenesis. In fact, the increased frequency of T regulatory cells (Tregs) in the colonic lamina propria of IBD patients is consistently lower than that observed in other inflammatory conditions such as infectious enteritis and diverticulitis [23]. Recently, a new impetus for investigation of the adaptive compartment of the immune system has been stimulated by discovery of the IL23/IL17 pathway and Th17 lymphocytes and their relevance to the inflammatory response in CD. This has led to the subsequent development of a novel mechanism of action for new drugs (i.e., ustekinumab) [24,25,26]. More recently, the possible role for CD onset as a consequence of a defect in innate immunity was proposed, depicting a scenario where a primarily immunodeficient condition (in innate immunity)—rather than an over-reactive immune response (in adaptive immunity)—appears to be the most relevant event. A fundamental role for the homeostatic maintenance in the balance between luminal contents (bacterial burden) and mucosal immunity is the intestinal barrier function of epithelial cells, including intestinal permeability and production of antibacterial molecules [27,28]. Alteration of such mechanisms are indicated as an early initiating event in the pathogenesis of CD [29]. Several studies indicate increased intestinal permeability as an early event in the onset of IBD [30,31]. A recent study showed that in the colonic mucosa of IBD patients, a positional remodeling of goblet cells with downregulation of WFDC2—an anti-protease molecule that inhibits bacterial growth—preserves the integrity of tight junctions and prevents invasion by commensal bacteria and mucosal inflammation [32]. Active secretion of mucin and defensins by epithelial cells has been found to be impaired in IBD patients [33,34]. The possible role of altered defensin production has been further confirmed by the detection of such defects in patients with polymorphisms of the CD-associated genes, nucleotide oligomerization domain 2 (NOD2), and autophagy related protein 16-like 1 (ATG16L1) [35,36]. Similar to other pathologic conditions, CD has recently undergone a so-called “microbioma revolution”, triggered by the new next-generation sequencing methods that have greatly increased research into host-bacteria interactions and allowed a consistent expansion of studies on the complexity of the ecosystem in the digestive system [37]. Besides great promise, intense research has produced a much greater amount of experimental, rather than clinical, data. Initial findings indicating a negative role of commensal bacteria on IBD onset —supported by experimental and clinical observations that a reduction of bacteria content (germ-free animals, fecal diversion) may attenuate or prevent inflammation [38,39]—has been replaced by a more modern vision. That vision indicates that reduction in diversity and an imbalance between certain bacterial species (“dysbiotic” microbiota) may contribute to the initiation and maintenance of chronic intestinal inflammation [40,41]. Despite the fact that single specific etiologic microorganisms for the pathogenesis of disease have not been identified, an increased Enterobacteriaceae/Firmicutes ratio has been observed in IBD patients vs. normal controls [42], as well as a reduction in possible protective species (i.e., Clostridial cluster IV and XIV, Bacteroides fragilis and Faecalibacterium prausnitzii) [43,44,45]. An increment of “enteropathogens” with proinflammatory properties (such as mucosa-adherent-invasive E. coli (AIEC) strains, pathogenic Yersinia and Listeria monocytogenes) has also been reported [46].

The increase in knowledge of CD pathogenesis has led to a clearer understanding of the complexity of the disease. Genetic predisposition, dysregulation of innate and adaptive immunity, microbiota alterations, and possible environmental factors all interplay, resulting in a number of clinical pictures that falls under the umbrella of Crohn’s disease. The simplified schematic view of CD in the past has been replaced by a modern vision in which multiple molecular pathways, redundancy, plasticity, and dynamicity concur to a different degree in the initiation, development, and maintenance of the condition. In line with this new outlook, different cell types and their mediators may have diverse (and even opposing) roles in the various disease stages and in different topographic compartments. Tumor necrosis factor (TNF) is a perfect example of a molecular mediator with multiple, and still not completely discovered, effects at the intestinal mucosa with potentially relevant roles in maintenance of homeostasis and inflammatory disease occurrence. Aim of the present review was to analyze the role of TNF in the onset and the maintenance of the inflammatory process characteristic of CD, and to describe the clinical application of the block of TNF, depicting the currently available drugs and the potential future implications.

## 2. TNF in The Pathogenesis of Crohn’s Disease: A Dichotomous Role

TNF is a homotrimer of 157 amino acid subunits, primarily produced in soluble or transmembrane form by activated macrophages, although other immune cells, such as dendritic cells, T cells, B cells, and mast cells—but also epithelial and endothelial cells and fibroblasts—can be induced to express TNF [47]. TNF exerts its functions by the interaction with two membrane receptors, TNFR1 and TNFR2. The first is expressed by every type of cell, has a cytoplasmic death domain, and probably mediates most of the biologic effects of TNF. After binding of soluble or membrane-bound TNF, the inhibitory protein silencer of death domains (SODD) is released from TNFR1′s intracellular domain (ICD). It is then recognized by the adaptor protein TNF receptor–associated death domain (TRADD), which in turn recruits additional adaptor proteins such as receptor-interacting protein (RIP), TNFR-associated factor 2 (TRAF2), and Fas-associated death domain (FADD). Different pathways are activated, resulting in apoptosis induction by caspases upon the initial cleavage of caspase 8 and the activation of two major transcription factors such as c-Jun and NF-κB. These transcription factors regulate a large number of genes involved in proliferation, inflammation, and immune response. TNFR2 expression is limited to endothelial, hematopoietic, and immune cells, and after binding with TNF, it recruits TRAF2 and initiates a molecular cascade that finally results in kinase and NF-κB activation. Extensive crosstalk exists among the different pathways stimulated by TNF at the cellular level, and multiple biological effects may result in the stimulation of different cell types by TNF [48]. In addition, the pleiotropic effect on apoptosis regulation by TNF may be related to the crosstalk between this molecule and other receptors, such as receptor tyrosine kinases (RTKs) [49], or other pathways, such as the Ras-association domain family member (RASSF)-dependent activation of Hippo pathway [50].

### 2.1. The Classical Pro-Inflammatory Effect of TNF

TNF was identified in 1975 and purified and cloned in 1985 [51,52]. Initial research focused on the potential clinical applications associated with anti-neoplastic effects observed in pre-clinical studies. However, clinical trials subsequently showed disappointing results for lack of clinical benefits and high toxicity. Afterwards, the observation of increased TNF in serum and feces of IBD patients, and the characterization of experimental models of TNF-dependent systemic and intestinal inflammation opened the door to identifying the pro-inflammatory effects of TNF and its potential relevance in IBD pathogenesis [53]. During inflammation, TNF is a crucial mediator in the late phase of the inflammatory cascade. TNF is also responsible for the maintenance and chronicity of intestinal mucosal inflammation through recruitment and activation of lymphocytes and granulocytes, local expression of adhesion molecules on endothelial cells at the site of inflammation, secretion of pro-inflammatory mediators (IFNγ and IL-6) and radical oxygen species, and the formation of edema and granulomata [54]. The relevant pro-inflammatory role of TNF in IBD is definitively confirmed by the impact that the drugs blocking TNF have had on the management of IBD patients, not only for symptom relief, but for the effects on more consistent outcomes of disease such as mucosal healing and reduced surgery [55].

### 2.2. Potential Protective and Anti-Inflammatory Effect of TNF

Aside from its proinflammatory function, consistent data exist regarding the role of TNF in mucosal homeostasis. TNF represents a fundamental cytokine of innate immunity that has an important role in maintaining the homeostatic balance between luminal contents (i.e., microbiota, pathogenic bacteria) and the mucosal immune system, actively participating in intestinal barrier function. In an experimental model of intestinal inflammation, such as dextran sulfate sodium (DSS)-induced colitis, an incremental increase in colitis severity was observed in mice after monoclonal anti-TNF administration [56,57]. Additionally, immunodeficient mice (RAG knockout (KO)) that lacked expression of TNFR1 had higher mortality and impaired epithelial regeneration [58]. Similar results were observed in another model of experimental colitis (2,4,6-Trinitrobenzenesulfonic acid (TNBS)) [59]. The exact molecular pathways that mediate the protective effects of TNF on the intestinal mucosa are not completely clear, but several hypotheses can be considered. In particular, the different anti- vs. pro-inflammatory effects of TNF could be caused by different cellular sources and/or mucosal site/targets of TNF, different responses elicited, and the direct effects on enhancing intestinal barrier integrity. Most of the TNF involved in the inflammatory cascade is produced by activated macrophages and lymphocytes, whereas smaller amounts can be produced by epithelial cells, and that may be partly directed against luminal bacteria and partly stimulate the enhancement of barrier defense by means of reducing permeability and stimulating protective factors production, such as chemokines (i.e., defensins). In a spontaneous mouse model closely resembling human CD, the SAMP1/YitFc (SAMP) model of CD-like ileitis, administration of a multispecies compound of probiotics at a high dose before the onset of intestinal disease effectively prevented intestinal inflammation by stimulation of epithelial TNF and a consequent permeability decrease, and interestingly, the preventive effect was abrogated by the administration of recombinant anti-TNF [60]. This finding was further confirmed by in vitro experimentation indicating that pre-treatment of ilea from pre-inflamed SAMP mice with probiotic-conditioned media or TNF decreased ileal paracellular permeability by modulating tight junction proteins [61]. The relevance of TNF and other innate cytokines in regard to intestinal homeostasis has also been confirmed by the link between the pattern recognition receptors (PRR) pathway and production of such molecules, mediated by the MyD88 signaling adaptor protein. In line with these findings, mice deficient in MyD88 had more severe colitis and higher morbidity after DSS administration, due to the lack of upregulation of innate cytokines, including TNF and IL-1β [62]. Another mechanism by which TNF may exert its protective role within the intestinal mucosa is by limiting the progression of inflammation through the stimulation of apoptosis of effector immunocytes within the lamina propria that have important roles in eliminating intruding luminal bacteria [63]. Moreover, TNF has been shown to stimulate the production of mucosal glucocorticosteroids with anti-inflammatory activity at mucosal sites after acute injury [64]. Recently, a TNFR2-mediated regulatory effect of TNF on the expression of IL17 in Treg lymphocytes was shown, suggesting a potential reciprocal regulation between these two cytokines [65,66]. Although consistent crosstalk and overlapping functions exist between TNFR1 and TNFR2, the latter is likely to electively mediate homeostatic effects such as cell proliferation/survival and tissue regeneration [67].

### 2.3. Potential Clinical Implications

Taken together, those findings suggest a definite distinct role of TNF in the early vs. late phase of the inflammatory process leading to Crohn’s disease [68]. In the early phase, as a breach in the innate immune defense and of the intestinal barrier seems to be the relevant momentum for the initiation of the inflammatory cascade, TNF may contribute to straighten the innate response in that crucial phase. In the late stage, TNF substantially contribute to the maintenance of the chronic inflammation characteristic of the active state of the disease (Figure 1). Drugs blocking TNF in that phase of the disease may consistently reduce the inflammatory state, with evident clinical short- and long-term benefits that will be discussed in the next section. Nonetheless, since many patients do not respond to anti-TNF therapy, it could be possible that in those patients, the inhibition of protective role of TNF prevails over the anti-inflammatory effect consequent to its block. Another potential implication of the dual role of anti-TNF in the different disease phases regards the appropriate timing of the pharmacological block of the TNF. Early intervention is generally associated with a higher response rate, but the positive pre-clinical role of the TNF in the maintenance of mucosal homeostasis may be considered for the future design of clinical trials, considering with high caution very early or even preventive treatments involving TNF block. Indeed, the growing clinical data (which will be further discussed in the next section) indicating a better clinical efficacy of anti-TNF in early vs. long-standing CD mainly refers to the time from the diagnosis of the disease, which probably already corresponds to the late phase of the pathogenetic inflammatory process. Conversely, the early inflammatory phase, where TNF may exert a protective role, is essentially pre-clinical, whereas symptoms (that may lead to the diagnosis) occur only when the inflammatory process is fully established. In patients with long-lasting CD, the inflammatory burden tends to be less prominent, while fibro-stenotic complications increase. Notwithstanding the pro-fibrotic effect of TNF [69], a potential preventive role of anti-TNF drugs on strictures in CD patients is not demonstrated, so that a more consistent effect of that drugs is observed in early uncomplicated disease. Moreover, since pro-inflammatory cytokines’ cascade includes many overlapping and interacting pathways, TNF appears to have a regulatory role for other pro-inflammatory mediators that may be overstimulated after its block. Therefore, in particular for complicated patients, the increased comprehension of such molecular pathways may lead to the development of future therapeutic strategies implying multiple simultaneous pro-inflammatory pathways block.

## 3. The TNF Block as a Therapeutic Target in Crohn’s Disease: Current Drugs Available

There are currently three anti-TNF drugs approved by the US Food and Drug Administration (FDA) for the treatment of CD patients: infliximab, adalimumab, and certolizumab pegol. Certolizumab pegol is not approved by European regulatory agency (European Medicine Agency—EMA) [70]. 

Infliximab is a chimeric IgG1 monoclonal antibody against TNF, and it was the first anti-TNF drug approved by FDA for the treatment of CD (in 1998, and by EMA in 1999), first demonstrating efficacy in 1997 [71]. It can fix, complement, and lyse cells expressing membrane-bound anti-TNF with a potent anti-inflammatory effect at the intestinal mucosal site involving probably more unknown pathways. It is administered in intravenous infusions at weeks 0, 2, and 6 (induction) and then every 8 weeks (maintenance). Pivotal trials demonstrating its efficacy in CD patients are the ACCENT I [72] and II [73], as well as the SONIC trial [74], but more trials and real-life studies further confirmed drug efficacy and safety.

Adalimumab is a human IgG1 anti-TNF antibody that was approved for clinical use in CD patients by the FDA in 2002 and by the EMA in 2003. It has the ability to fix complement and lyse cell expressing TNF, and it is administered by subcutaneous injection every other week. Among many studies demonstrating efficacy and safety of that drug in CD patients, the CLASSIC I and II [75,76] trials demonstrated efficacy in induction and maintenance of remission, while CHARM [77], GAIN [78], and Extend [79] trials proved the efficacy of the drug in fistulizing disease, as a second line after infliximab failure, and for mucosal healing achievement.

Certolizumab pegol is a PEGylated Fab’ fragment of a humanized monoclonal anti-TNF antibody with a high binding affinity for soluble and transmembrane TNF and less immunogenicity side effects due to the lack of the Fc portion. It does not induce apoptosis nor fix complement. It is administered subcutaneously and, having a longer half-life than adalimumab due to the PEG addition, has a maintenance dosing every month. The registrative trials PRECISE 1 [80], PRECISE 2 [81], and PRECISE 3 [82] demonstrated the efficacy of the drug in CD patients, but relevant endpoints, such as induction of remission and response rate over placebo at week 6 and 26, were not achieved. The drug is not approved for clinical utilization for CD by the EMA, but it was approved by the FDA in 2008 for the treatment of signs and symptoms in CD patients who fail to respond to other conventional treatments.

Data of the main clinical trials of the three anti-TNF agents currently used in CD patients are summarized in Table 1. Those drugs, which so profoundly revolutionized the therapeutic approach in IBD patients, still represent the cornerstone of treatment for moderate–severe CD. The efficacy of anti-TNF in inducing and maintaining of response and remission in CD patients has been further confirmed by a network meta-analysis including 10 studies, with no clear superiority among drugs [83]. Indeed, besides the efficacy for inflammatory luminal disease, such therapies have been proven beneficial in specific complex clinical situations, such as prevention of post-operative recurrence [84], peri-anal fistulizing disease [85], concurrent extra-intestinal manifestation [86], and small bowel stenosis [87]. Therefore, notwithstanding the novel drugs already available, anti-TNF agents (infliximab, adalimumab, and certolizumab) still represent indispensable tools in the therapeutic armamentarium for CD management.

Other anti-TNF drugs did not show efficacy in RCTs in CD patients including: etanercept [88], a soluble form of the p75 receptor that inhibits TNF, already approved for rheumatologic and dermatologic indications; CDP571 [89], an engineered human monoclonal antibody against TNF; and onercept [90], a recombinant soluble human p55 receptor to TNF. The reasons for the lack of efficacy of such anti-TNF agents are not completely clear: CDP571 is an IgG4 antibody, while infliximab and adalimumab are IgG1, and such a difference may lead to some alterations that may be relevant for the clinical actions. For etanercept and onercept, the crucial issue is probably the fact that they bind only the soluble form of TNF, which is most likely not sufficient to determine clinical effects in CD. Moreover, since etanercept is largely used in rheumatology and dermatology, consistent risk for new onset or exacerbation of IBD has been described [91,92]. A possible explanation is that, as etanercept is a soluble TNFRII receptor, it binds not only the TNF but even other ligands such as lymphotoxin-α, which is a relevant cytokine for mucosal homeostasis [93]. Besides etanercept, the possible onset of IBD in patients treated with others anti-TNF for non-gastroenterological indications is rarely reported [91], further reflecting the complexity and the redundancy of the TNF pathway, and the possibility that, in a subset of patients, the block of TNF may worsen the inflammatory process.

Despite the widespread use of anti-TNF agents since the last two decades, some open issues are still up for debate. According to clinical trials and real-life studies, about 20% of patients do not respond to anti-TNF ab initio (primary failure), and about 20–30% of patients are likely to lose response every year (secondary failure). Therefore, a consistent portion of patients would need a second-line therapy or a surgical approach. In order to increase efficacy of therapy and to ideally affect the natural course of the disease, “early treatment” has been proposed—by 2 years from the initial diagnosis—in selected CD patients with features of severe disease (i.e., young age, extensive intestinal involvement, perianal disease, smokers). In the SUTD (Step Up vs. Top Down) trial, an early immunosuppression plus infliximab administration was associated with a higher clinical remission rate at 2 years in 133 CD patients compared to patients with conventional management [94]. In the REACT trial, in a larger set of patients (more than 1700) comparing early immunosuppression plus adalimumab vs. conventional management, only a trend for a higher rate of clinical remission at 2 years was found in the early treatment group, but when more consistent outcomes were considered (hospital admission, surgery, and serious disease-related complications), a significant difference emerged [95].

The observation that anti-drug antibodies are associated with infusion reactions and loss of response—and that therapeutic efficacy is coupled with adequate blood drug level—has pushed toward the concept that the measurement of such parameters (i.e., anti-TNF Ab and drug trough levels) would optimize anti-TNF drug utilization [96]. Therapeutic drug monitoring (TDM) can be used in two main clinical settings: in primary and secondary biologic failure patients in order to more appropriately assess consequent therapeutic strategy such as optimization, switch-in-class, switch-out-of-class (reactive TDM) or in patients in remission to drive to a tailored approach to drug administration to maintain efficacy and minimize side-effects (pro-active TDM) [97]. Despite the rational and the cost-saving effect, no decisive evidence is available for a significant superiority of TDM over the sole clinical approach both for the reactive and pro-active monitoring. Yanai et al. demonstrated in a retrospective study that TDM can help in the management of patients who lost response to infliximab and adalimumab [98]. A Danish cohort study indicated that TDM-guided management had a similar clinical outcome at 12 weeks as an empirical approach, but was cost-effective [99], and some studies confirmed that finding [100,101]. For pro-active TDM, even less evidence is available, and the two main trials addressing the potential utility of that approach; i.e., the TAXIT and TAILORIX trials failed to show significant benefits over the clinical based management [102,103]. Considering that—and the fact that controversies exist for assay methods and reference levels—TDM is generally not routinely performed in clinical practice, but it can be useful in the management of anti-TNF failure patients.

Despite anti-TNF drugs are generally considered safe, adverse events are still possible and some pre-existing clinical condition may contraindicate therapy or require special attention. In fact, drug- (infusion reactions or local reactions in the site of injection), class- (infections, malignancy, cardiac failure, neurologic disease, paradoxical rheumatologic and dermatologic inflammation), and disease-specific (intestinal inflammatory flare, stenosis, abscesses) adverse events may occur, sometimes leading to discontinuation of therapy and/or hospitalization [104]. Therefore, anti-TNF drugs are contraindicated in patients with congestive heart failure of III/IV New York Heart Association (NYHA) class and active demyelinating neurologic disease. Anti-TNF drugs should be given with great caution to elderly patients, to those at higher infective risk, and to those with a recent history of neoplastic disease. Novel biologic drugs with different mechanism of action (i.e., Ab anti-integrin α4β7 and anti-IL12/23) appear to have a more favorable safety profile, and they can be probably used in such high-risk patients with more ease. Most of the safety data for such novel drugs come from RCT, and real-life and long-term post-marketing data are scarcer than that for anti-TNF, so the promising higher safety needs to be further evaluated over time.

## 4. Discussion

TNF is one of the most extensively investigated molecular mediators in CD patients. The clinical application of such intense research is rendered evident with the development of the anti-TNF class of biologic drugs, which presently represent the main reference for therapy of moderate–severe disease not responding to conventional therapy. As the comprehension of the pathogenesis of the disease increases, novel potential mechanisms of action with a possible role in the onset and maintenance of the chronic mucosal inflammation are investigated, eventually leading to the proposal of novel drugs for the management of CD patients. Anti-TNF block do not represent the exclusive mechanism of action of biologic drugs in CD, but therapeutic approach targeting the leucocyte homing and the IL12-23 pathway are already available in the market, and more drugs with new mechanisms of action are ready to be proposed or are being investigated in clinical trials [14]. In order to incorporate those new therapeutic acquisitions into a wider and prospective vision, it is of relevant importance that the clinician maintains a balanced and independent view on the available evidence about therapy. The modern IBD specialist has the uneasy role of adequately considering and weighing the different evidence coming from guidelines, meta-analysis, clinical trials, and translating it into the daily clinical practice, where usually the single specific patients need a tailored approach. From one side, novel therapeutic drugs offer fascinating new opportunities of treatment and promise to overcome unfilled needs of previous therapies. Such new drugs are usually supported by a strong marketing endorsement and positive results coming from registrative trials may be overemphasized. Conversely, since new drugs are generally more costly, in some Countries the health administrations can retain and limit to a variable degree the free utilization of those drugs in order to reduce the health expenses. This is further stressed by the outbreak on the market of the biosimilars that has, in a few years, consistently pushed down the expenses for biological anti-TNF therapy. They have the potential to treat a larger number of patients at cost parity and to extend availability of biologics in countries where those drugs were unaffordable for few years beforehand.

Besides classical pro-inflammatory role in the onset and maintenance of chronic intestinal inflammation, more recent studies have highlighted a possible protective role of TNFα at mucosal level. This finding from one side may confirm the high complexity of the homeostatic mechanism of the intestinal mucosa, in which multiple pathways may reciprocally interact and contribute in turn either to the protection of the mucosa or to its alteration by unregulated inflammation. On the other hand, the perception of the multiple and even opposite effect that single molecular mediators may provoke opens the door to the implementation and the refining of the positioning of drugs interfering with that specific molecular pattern. Thus, both for clinical/economic reasons and for the growing comprehension of the multiple molecular function at mucosal level, discussing “potential for intervention” for TNF in CD is not anachronistic. In fact, as the basic and translational research progress, a new impulse and novel possible implications may arise even for a molecular mechanism that has already been investigated for a long time, such as TNF. To further confirm that, the National Institute of Health (NIH) database (ClincalTrails.gov) presently has a total of 220 trials involving anti-TNF in CD patients registered, among which 173 are completed (85 for infliximab, 56 for adalimumab, and 32 for certolizumab) and 47 are still recruiting patients (24 for infliximab, 19 for adalimumab, and 4 for certolizumab).

As we just entered a new decade, we can imagine a future scenario in which more novel drugs will become available for the treatment of CD patients and more biological drugs’ patents will expire, further expanding therapeutic medical options and therapy availability. Key points for the future developments of positioning and implementation of TNF-based therapeutic strategies would involve clinical and basic research. For clinical research, the design and implementation of fair, well-designed, head-to-head clinical trials will help clinicians to orientate among the increased therapeutic options available. To date, only the Varsity study, in UC patients, directly compared two different biologic drugs in the same set of patients [105], but more head-to-head studies are coming in the near future. In basic and translational research, the better comprehension of the molecular mediators involved in the different pathways would push toward the identification of clinical features and/or molecular markers predictive of response of a specific drug in the single patients. Various clinical features, such as phenotype of disease, young age, isolated colitis, and smoking, have been investigated as potential predictors of response to anti-TNF therapy, but with inconclusive results [106]. Very recently a large prospective cohort study including 955 infliximab-treated and 655 adalimumab-treated anti-TNF naïve patients with active luminal CD [107] investigated possible predictors of therapy failure: at multivariate analysis low drug concentration at week 14 was the only factor independently associated for both drugs with therapy failure at week 14 and 54, while smoking and obesity were significantly associated with treatment failure at week 54 only for adalimumab. Considering the gene expression, a five gene set (i.e., TNFAIP6, S100A8, IL11, G0S2, S100A9) has demonstrated its ability to accurately predict the response to infliximab in patients with CD colitis [108]. Moreover, few studies have indicated the potential utilization of single photon emission computed tomography (SPECT) and confocal microscopy with fluorescent antibodies to TNF during colonoscopy to predict the response to anti-TNF therapy [109,110]. Recently, two studies investigated the occurrence of specific inflammatory pathways in IBD patients associated with a low response to anti-TNF therapy. First, West et al. observed high tissue levels of oncostatin M in an animal model of anti-TNF-resistant intestinal inflammation and in anti-TNF non-responder IBD patients [111]. Second, Gaujoux et al. demonstrated a significant plasma cell increase in biopsies from anti-TNF non-responder patients in conjunction with an increase in triggering receptor expressed on myeloid cells 1 (TREM-1) and the chemokine receptor type 2 (CCR2)–chemokine ligand 7 (CCL7) axes. This suggests potential utilization of systemic TREM-1 expression as a non-invasive diagnostic marker of non-response to antiTNF therapy at baseline [112]. Moreover, microbiome composition difference prior to treatment has been preliminarily investigated as a potential predictor marker for response to infliximab [113]. At present, no single marker can be proposed as an appropriate prognostic indicator for response to any biologic drug [114]. The characterization of molecular features associated with different responses would help in the future in guiding the selection of patients to a specific therapy, with a relevant increase in therapy efficacy and ideally avoiding multiple empirical therapy lines.

In addition, a higher definition of the multiple molecular interactions in CD may lead to the design and development of pre-clinical and clinical studies including multiple biologics association. At present, few case reports are available for association therapy of biologic drugs in CD [115,116,117,118], and a single clinical trial by Sands et al. in 2007 preliminarily evaluated the safety of infliximab plus natalizumab in 79 refractive patients [119]. As in the case of oncologic therapy, a therapeutic approach towards multiple molecular pathways would offer higher chances of short- and long-term therapeutic efficacy in particular for CD patients with complex refractory disease, but the safety issue needs to be carefully evaluated. Besides efficacy increasing, a crucial issue of the present and future therapeutic approach is represented by the safety: a more and more appropriate and mindful patient selection to therapy will ideally lead to a consistent reduction of mis- and over-treatment with the final goal of tailoring a specific therapy to the single patient.

## 5. Conclusions

Despite consistent research and some advance in the comprehension of IBD, many aspects still remain insufficiently known—in particular, the disease pathogenesis. The more our knowledge seems to shed light in this obscure cave, the more novel questions arise, making more puzzling, and still fascinating, our path toward the truth. CD appears more and more as a complex condition where multiple different factors, reciprocally interacting, contribute to the variable clinical phenotype of disease that we face in our clinical daily practice. Still, some aspect of this complex puzzle has been clarified, leading to important discoveries in the therapeutic field. This is the case of biologic drugs that, interfering with specific molecular pathways of the inflammatory cascade, has dramatically expanded our therapeutic impact on the disease and profoundly changed the management approach to CD patients. Among those, the anti-TNF agents have been the first drugs discovered and remain the standard of therapy for moderate–severe disease patients who did not respond to the conventional therapy. The consistent efficacy of those drugs straightens the important role of the TNF for the disease maintenance in CD. Nonetheless, in line with the complexity of the disease, the same TNF has multiple complex molecular effects and potentially different roles in the initiation and maintenance of the mucosal chronic inflammation. The further progress of the research in IBD would hopefully help to better unravel the multiple complex role of TNF and of other molecular mediators. The desirable final result is to implement and optimize the current available drugs utilization as well as the future development of novel drugs and strategies.

## Figures and Tables

**Figure 1 ijms-22-10273-f001:**
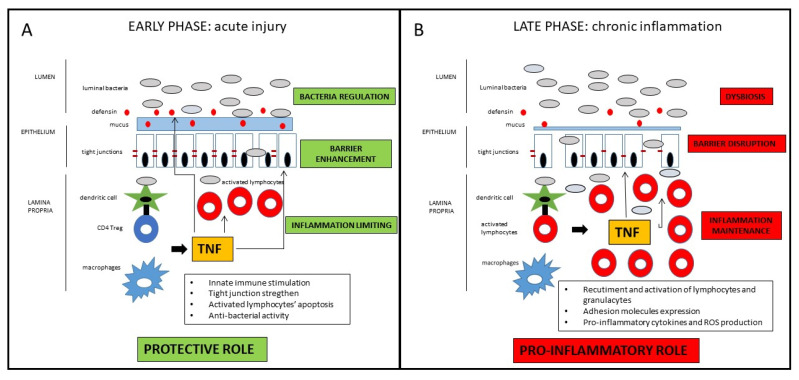
Schematic representation of the dichotomous role of TNF in CD pathogenesis. In the early phase of the inflammatory process, TNF has protective effects on the intestinal mucosa, whereas in the late phase of disease, TNF has consistent pro-inflammatory effects. For detailed description, please refer to the text.

**Table 1 ijms-22-10273-t001:** Pivotal clinical trials of the three anti-TNF drugs approved for CD patients.

Drug	Trials (Ref)	Patients	End-Points	Results
*Infliximab (IFX)*	ACCENT I (72) ACCENT II (73) SONIC (74)	335/573 responders at week 2 282 222	Remission at week 30 Fistulas’ healing at week 54 Steroid-free remission and mucosal healing at week 26	IFX 5 mg/Kg: 44/113 (39%) IFX 10 mg/Kg: 50/112 (45%) Placebo: 23/110 (21%) IFX induction + 8 week maintenance: 50/138 (36%) IFX induction + placebo: 27/144 (19%) Remission—IFX + AZA: 96/169 (57%) IFX: 75/169 (44%) AZA: 51/170 (30%) Mucosal healing—IFX + AZA: 47/107 (44%) IFX: 28/93 (30%) AZA 18/109 (17%)
*Adalimumab (ADA)*	CLASSIC I (75) CLASSIC II(76) CHARM (77) GAIN (78) Extend (79)	299 259 499/854 responders at week 4 325 135	Remission at week 4 Remission at week 56 Remission at week 26 and 56 Fistulas healing at week 56 Remission rate at week 4 in patients IFX non responders Mucosal healing at week 12 and 52	ADA 160/80 mg: 27/76 (36%) ADA 80/40 mg: 18/75 (24%) ADA 40/20 mg: 13/74 (18%) Placebo: 9/74 (12%) ADA 40 mg/2 week: 15/19 (79%) ADA 40 mg/week: 15/18 (83%) Placebo: 8/18 (44%) ADA with dose optimization: 93/204 (46%) Week 26—ADA 40 mg/2 week: 68/172 (40%) ADA 40 mg/week: 75/157 (47%) Placebo: 29/170 (17%) Week 56—ADA 40 mg/2 week: 62/172 (36%) ADA 40 mg/week: 65/157 (41%) Placebo: 20/170 (12%) ADA 40 mg/2 week: 10/30 (33%) ADA 40 mg/week: 11/40 (28%) Placebo: 6/47 (13%) ADA: 34/159 (21%) Placebo: 12/166 (7%) Week 12—ADA 40 mg/2 week: 17/62 (27%) Placebo: 8/61 (13%) Week 52—ADA 40 mg/2 week: 15/62 (24%) Placebo: 0/61 (0%)
*Certolizumab pegol (CZP)*	PRECISE1 (80) PRECISE2 (81) PRECISE3 (82)	655 425 241 from PRECISE2	Remission at week 6 and 26 Remission at week 26 Remission at week 52 and 80	Week 6—CZP: 71/329 (22%) Placebo: 57/326 (17%) Week 6 and 26—CZP: 47/327 (14%) Placebo: 32/326 (10%) CZP: 103/215 (48%) Placebo: 61/210 (29%) Week 52—CZP continuous group: 58/141 (41%) Drug-interruption group *: 30/100 (30%) Week 80—CZP continuous group: 51/141 (36%) Drug-interruption group *: 23/100 (23%)

* Patients who had CZP from week 0 to 6, then placebo from week 6 to 26 (from PRECISE2), then CZP to week 80 (in PRECISE3).

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
