# Peer review of "Tumor Necrosis Factor’s Pathway in Crohn’s Disease: Potential for Intervention"

_ijms, 2021, doi:10.3390/ijms221910273_

Round 1
Reviewer 1 Report
The paper will be ready for publication after light revisions.
The review is well designed with a correct sequence in the illustration of the data.
However, some my minor, below-mentioned, concerns should be addressed by the authors:
- Why didn't the authors mention some studies like the EFIFECT study or the WELCOME study?
- Authors should report more recent references
- In my opinion, authors should also analyze recently published meta-analysis to avoid neglecting previous reviews on the same topic
Author Response
We thank the reviewer for the positive comments on our paper, and we addressed his/her suggestions. In particular:
- We limited the report of the clinical trials to the main ones (i.e. registrative trials or trials with results of particular interest). For the purpose of the review we did not intend to make a systematic review of the trials, but rather to report the finding of the main ones and highlight and discuss some specific topics related to anti-TNF therapy.
- References has been consistently revised and updated with more recent papers
- We added recent meta-analysis about the clinical efficacy of anti-TNF drugs in CD patients as suggested
Reviewer 2 Report
The review may give the insights to implement and optimize the utilization of the currently available drug. I have some comments on this manuscript.
Introduction
For the introduction section, the author should state the purpose of this review.
Too many details are described. Only a brief explanation of medication or diagnosis for CD is required.
(Line 78-)
Dividing the paragraph for the sentences about “biologics” is recommended.
(Lines 279-)
Please divide the section for each biologic agent to read easily.
(Lines 385-390)
Please add references for these sentences
(Lines 417-)
The reviewer is not certain whether the issues on “the health expenses” are common for the IBD patients because they need to cover all the expenses for the treatment.
(a designated intractable disease?)
(Line 496)
“remain unraveled” means “not to be unrevealed”?
Author Response
We thank the reviewer for the comments on our paper, and we tried to accomplish all the suggested items. In particular:
- The purpose of the review has been stated at the end of the Introduction section
- The Introduction has been shortened
- We divided the paragraph in the sentence about “biologic”
- We divided the section for each biologic agent as suggested
- Since the issue of the health expenses may vary in many Countries, we added “in some Countries” to the sentence
- “remain unraveled” has been corrected to “remain still insufficiently known”
Reviewer 3 Report
- Please add the description of the crosstalk between TNF and other receptors.
- Please write the full name for the abbreviation at the first time in this manuscript. For example: L230: DSS.
Author Response
We thank the reviewer for the positive evaluation of our paper.
- We double-check the abbreviations and we put the full name at the first appearance
- We added a sentence at the end of the “TNF in the pathogenesis of Crohn’s disease: a dichotomous role” section about the possible crosstalk of TNF. We did not discuss in deep the subject in order to remain in the purpose of the present review.